# Targeting with Structural Analogs of Natural Products the Purine Salvage Pathway in *Leishmania (Leishmania) infantum* by Computer-Aided Drug-Design Approaches

**DOI:** 10.3390/tropicalmed9020041

**Published:** 2024-02-03

**Authors:** Haruna Luz Barazorda-Ccahuana, Eymi Gladys Cárcamo-Rodriguez, Angela Emperatriz Centeno-Lopez, Alexsandro Sobreira Galdino, Ricardo Andrez Machado-de-Ávila, Rodolfo Cordeiro Giunchetti, Eduardo Antonio Ferraz Coelho, Miguel Angel Chávez-Fumagalli

**Affiliations:** 1Computational Biology and Chemistry Research Group, Vicerrectorado de Investigación, Universidad Católica de Santa María, Arequipa 04000, Peru; hbarazorda@ucsm.edu.pe (H.L.B.-C.); 75559037@ucsm.edu.pe (E.G.C.-R.); angela.centeno@ucsm.edu.pe (A.E.C.-L.); 2Facultad de Ciencias Farmacéuticas, Bioquímicas y Biotecnológicas, Universidad Católica de Santa María, Arequipa 04000, Peru; 3Laboratório de Biotecnologia de Microrganismos, Universidade Federal São João Del-Rei, Divinópolis 35501-296, MG, Brazil; asgaldino@ufsj.edu.br; 4Programa de Pós-Graduação em Ciências da Saúde, Universidade do Extremo Sul Catarinense, Criciúma 88806-000, SC, Brazil; r_andrez@unesc.net; 5Laboratório de Biologia das Interações Celulares, Instituto de Ciências Biológicas, Universidade Federal de Minas Gerais, Belo Horizonte 31270-901, MG, Brazil; giunchetti@icb.ufmg.br; 6Instituto Nacional de Ciência e Tecnologia em Doenças Tropicais, INCT-DT, Salvador 40015-970, BA, Brazil; 7Programa de Pós-Graduação em Ciências da Saúde: Infectologia e Medicina Tropical, Faculdade de Medicina, Universidade Federal de Minas Gerais, Belo Horizonte 31270-901, MG, Brazil; eduardoferrazcoelho@yahoo.com.br; 8Departamento de Patologia Clínica, COLTEC, Universidade Federal de Minas Gerais, Belo Horizonte 31270-901, MG, Brazil

**Keywords:** drug discovery, natural products, Skimmianine, Visceral Leishmaniasis, molecular docking simulation, molecular dynamics simulation, virtual screening

## Abstract

Visceral Leishmaniasis (VL) has a high death rate, with 500,000 new cases and 50,000 deaths occurring annually. Despite the development of novel strategies and technologies, there is no adequate treatment for the disease. Therefore, the purpose of this study is to find structural analogs of natural products as potential novel drugs to treat VL. We selected structural analogs from natural products that have shown antileishmanial activities, and that may impede the purine salvage pathway using computer-aided drug-design (CADD) approaches. For these, we started with the vastly studied target in the pathway, the adenine phosphoribosyl transferase (APRT) protein, which alone is non-essential for the survival of the parasite. Keeping this in mind, we search for a substance that can bind to multiple targets throughout the pathway. Computational techniques were used to study the purine salvage pathway from *Leishmania infantum*, and molecular dynamic simulations were used to gather information on the interactions between ligands and proteins. Because of its low homology to human proteins and its essential role in the purine salvage pathway proteins network interaction, the findings further highlight the significance of adenylosuccinate lyase protein (ADL) as a therapeutic target. An analog of the alkaloid Skimmianine, N,N-diethyl-4-methoxy-1-benzofuran-6-carboxamide, demonstrated a good binding affinity to APRT and ADL targets, no expected toxicity, and potential for oral route administration. This study indicates that the compound may have antileishmanial activity, which was granted *in vitro* and *in vivo* experiments to settle this finding in the future.

## 1. Introduction

Computer-aided drug design (CADD) is a cutting-edge computational approach used in the drug development process, which is becoming embraced in both academic and pharmaceutical spheres [1]. CADD has been used to identify leads from chemical or natural compounds, and involves approaches like computational biology and chemistry, molecular modeling, and virtual screening [2], and has the potential to speed up the drug-design process and cut costs by several billion USD [3]. This approach shows particular potential for generating drugs to treat neglected tropical diseases (NTDs), a class of infectious diseases with high endemicity in developing nations that are deemed “neglected” when there are not any reliable, inexpensive, or simple-to-apply pharmacological treatments available [4]. Visceral leishmaniasis (VL) is one of the NTDs and ranks second and seventh among them in terms of mortality and loss of disability-adjusted life years, respectively [5,6]. VL is a vector-borne disease that can be fatal if untreated and is brought on by an intracellular protozoon of the genus Leishmania. The disease is caused by *Leishmania donovani* in Asia and Eastern Africa and *Leishmania infantum* in Latin America and the Mediterranean region [7]. With an estimated 200 million people at risk, VL is widespread in over 70 countries. However, seven nations—namely Brazil, Ethiopia, India, Kenya, Somalia, South Sudan, and Sudan—report more than 90% of all recorded cases of VL globally [8]. The only drugs authorized for the treatment of VL are pentavalent antimonial drugs, such as sodium stibogluconate and meglumine antimoniate, paromomycin, miltefosine, and amphotericin B [9]. However, these drugs are toxic to the liver, kidneys, and spleen and are still difficult to administer to patients [10]. Because of these factors, the search for novel treatments has recently been quite active, and a wide range of compounds have been discovered, yet few new drug candidates have reached clinical trials in recent decades [11].

Recent antileishmanial drug research has concentrated on natural products (NPs) and their potential [12,13,14,15,16,17], as there are 20 antiparasitic drugs derived from NPs in the total number of drugs introduced to the market over the past 40 years, which represents a significant source of new antiparasitic pharmacological entities [18]. Nonetheless, finding effective treatments for VL remains a major issue and requires the use of current technologies to find innovative chemotherapeutics. An important stage of drug development is the identification of appropriate therapeutic targets in the biological pathway, whereas two rules have been proposed to govern the identifying procedure. The first criterion is to identify a plausible target that is either missing from the host or physically and/or functionally distinct from the host homolog. Finding targets that are important for the parasite’s survival is the second one [19]. Additionally, single drugs that can function on several molecular targets are becoming more popular, and this appears like a viable strategy for treating complex disorders, such as VL [20]. By understanding the pathogen’s biological pathways—which requires knowledge of the underlying kinetics driving the enzymes and chemicals involved in the system—CADD approaches can be utilized, in this manner, to find novel promising substances that target a pathway’s principal enzymes [21,22].

Leishmania demonstrates its parasitism by extracting essential components from the host cells for their metabolic requirements, maintenance, and survival [23]; since the parasite lacks a de novo purine biosynthesis, it needs the host’s preformed purine nucleosides and bases to be salvaged to create new nucleoside monophosphates. Likewise, nucleosides can either be salvaged directly or first cleaved to release nucleobases, while cleavage can be performed by nucleoside hydrolases or by phosphorylases [24]. As a result, research on the purine salvage pathway (PSP) has drawn attention in the hopes of developing antileishmanial drugs [25]. The major enzymes of the pathway are phosphoribosyltransferases (PRTs), which convert dephosphorylated purines into the corresponding nucleoside monophosphates. Adenine phosphoribosyltransferase (APRT), hypoxanthine-guanine phosphoribosyltransferase (HGPRT), and xanthine phosphoribosyltransferase (XPRT) are the three PRTs that Leishmania expresses [26]. The parasite also expresses a significant number of purine interconversion enzymes, such as adenine aminohydrolase, which catalyzes the conversion of adenine to hypoxanthine, allowing it to survive with just one purine supply [27]. Recently, APRT has been used in molecular screening investigations against datasets of NPs obtained from actinomycetes [28], secondary metabolites from plants [29], and literature [30] for potential inhibitors. Likewise, a valuable treatment for canine visceral leishmaniasis (CVL) is allopurinol, a purine analog that is phosphorylated by HGPRT and incorporated into nucleic acids, killing the parasite [31], emphasizing the potential of PSP for drug discovery against Leishmania. However, as Leishmania contains several complementary purine salvage pathways, it has been suggested that PRTs are not essential for the parasite’s survival. Consequently, developing an antileishmanial treatment based on PSP requires focusing on multiple enzymes at once [32]. Finding a potential chemical candidate that could be utilized to treat VL was our main goal. To uncover compounds that can interact with various PSP targets, structural analogs of NPs that have demonstrated antileishmanial and anti-APRT properties were put through virtual screening. The natural products were retrieved from the NuBBEDB database, which provides pharmacological properties for each component based on literature descriptions [33]. Also, the PSP from *L. infantum* was studied using computational methods, and data on interactions between ligands and proteins was obtained by molecular dynamic simulations.

## 2. Methods

### 2.1. Phylogenetic Analysis

The FASTA sequence of the Adenine phosphoribosyltransferase (APRT) protein from *L. infantum* (ID: A4I1V1) was retrieved from the UniProt database (http://www.uniprot.org/) [34], and subjected to the BLASTp tool [35], followed by a sequence similarity search performed on *Leishmania braziliensis* species complex (taxid:37617), *Leishmania (Leishmania) amazonenses* (taxid:5659), *Leishmania panamensis* (taxid:5679), *Leishmania guyanensis* species complex (taxid:38579), *Leishmania mexicana* species complex (taxid:38582), and *Leishmania (Leishmania) infantum* (taxid:5671). To investigate the evolutionary relationship of Leishmanial APRT proteins, the retrieved sequences were uploaded in MegaX software and aligned using ClustalW [36]. Utilizing the neighbor-joining technique utilized by iTOL, the resulting multiple sequence alignment was used to reconstruct and show a distance-based phylogenetic tree [37].

### 2.2. Protein-Protein Interaction Network Analysis

The *L. infantum*-APRT sequence was analyzed for its interaction with other molecules from the purine salvage pathway. The molecular networks were retrieved from the STRING database (https://string-db.org) [38] and sent to the Cytoscape platform [39], where the plugin cytoHubba was used to score and rank the nodes according to network properties, affording to the Maximal Clique Centrality (MCC) topological analysis method [40]. To visualize the network, Cytoscape default settings were considered, whereas node size and color were manually adjusted, considering the scores provided by the MCC analysis.

### 2.3. Mining of Homologous to Human Proteins of the Purine Salvage Pathway

A search for sequence similarity was carried out on the human proteome database (*Homo sapiens* (taxid:9606)) using the BLASTp program [35] on the protein sequences belonging to the purine salvage pathway. When considering homologous sequences, an expected value (e-value) lower than 0.005 and a hit score larger than 100.0 were utilized. Proteins that displayed hits above the cut-off values were regarded as non-homologs [41,42]. Using the “circlize” package [43] in the R programming environment (version 4.0.3), chord plots were created to display the BLASTp scores for each protein [44].

### 2.4. Data Collection and Structural Analogs Search

The data collection strategy was adapted from [45], whereas the search for natural products with antileishmanial and APRT activity was performed at the Nuclei of Bioassays, Ecophysiology, and Biosynthesis of Natural Products Database (NuBBEDB) online web server (version 2017) (https://nubbe.iq.unesp.br/portal/nubbe-search.html, accessed on 15 April 2023), which contains the information of more than 2000 natural products and derivatives [33]; while the “antileishmanial property” was selected in the biological properties segment of the web server. The bibliographic data extraction, regarding the compounds found in NuBBEDB, was performed from the National Center for Biotechnology Information (NCBI) databases (https://www.ncbi.nlm.nih.gov/pubmed/, accessed on 20 May 2022); and the simplified molecular-input line-entry system (SMILEs) was searched and retrieved from PubChem server (https://pubchem.ncbi.nlm.nih.gov/, accessed on 23 May 2022) [46]. The SMILEs from the compounds were used for high throughput screening to investigate structural analogs by the SwissSimilarity server (http://www.swisssimilarity.ch/index.php, accessed on 27 May 2023) [47]; while the commercial class of compounds was selected and the zinc-drug-like compound library with the combined screening method was chosen for the high throughput screening to achieve the best structural analogs. Threshold values for positivity were selected by default parameters.

### 2.5. Molecular Properties Calculation

The Osiris DataWarrior v05.02.01 software [48] was employed to generate the dataset’s structure data files (SDFs). This followed the uploading to the Konstanz Information Miner (KNIME) Analytics Platform [49], where the “Lipinski’s Rule-of-Five” node was employed to calculate physicochemical properties of therapeutic interest—namely: molecular weight (MW), octanol/water partition coefficient (clogP), number of H-bond donor atoms (HBD) and number of H-bond acceptor atoms (HBA)—of the dataset. To generate a visual representation of the chemical space of the dataset for the auto-scaled properties of pharmaceutical interest, the principal component analysis (PCA), which reduces data dimensions by geometrically projecting them onto lower dimensions called principal components (PCs), calculated by the “PCA” KNIME node. Three-dimensional scatter-plot representations were generated for PCA with the Plotly KNIME node. Also, the Osiris DataWarrior software was employed to calculate the potential tumorigenic, mutagenic, reproductive effect, and irritant action of selected compounds predicted by comparison to a precompiled fragment library derived from the RTECS (Registry of Toxic Effects of Chemical Substances) database and to calculate the drug-likeness score of the compounds from the dataset; the calculation is based on a library of 5300 substructure fragments and their associated drug-likeness scores. This library was prepared by fragmenting 3300 commercial drugs as well as 15,000 commercial non-drug-like Fluka [48].

### 2.6. Virtual Screening

The FASTA sequences of the purine salvage pathway from *Leishmania infantum*—namely Adenine phosphoribosyltransferase (ID: A4I1V1), Adenylosuccinate lyase (ID: A4HS40), Putative AMP deaminase (ID: A4I876), AMP deaminase (ID: A4IC17), Guanine deaminase (ID: A4I4E1), GMP synthase (ID: E9AGZ1), Adenylate kinase isoenzyme (ID: A4I5L5), Guanylate kinase-like protein (ID: A4IDK0), Adenosine kinase (ID: A4I5C0), and Adenosine kinase (ID: A4IAC6)—were retrieved from the UniProt database (http://www.uniprot.org/), accessed on 3 June 2023), and subjected to automated modeling in SWISS-MODEL [50]. Furthermore, the compounds were imported into OpenBabel within the Python Prescription Virtual Screening Tool [51] and subjected to energy minimization. PyRx performs structure-based virtual screening by applying docking simulations using the AutoDock Vina tool [52]. The drug targets were uploaded as macromolecules, and a thorough search was carried out by enabling the “Run AutoGrid” option, which creates configuration files for the grid parameter’s lowest energy pose, and then the “Run AutoDock” option, which uses the Lamarckian GA docking algorithm. The entire set of modeled 3D models was used as the search space for the study. The docking simulation was then run with an exhaustiveness setting of 20 and instructed to produce only the lowest energy pose. The Z-score was calculated for each dataset, and the results were uploaded within the GraphPad Prism software version 10.0.2 (232) for Windows from GraphPad Software, San Diego, CA, USA, at http://www.graphpad.com. For the selected compounds, the Tanimoto similarity score was calculated for clustering the chosen compounds. The atom-pair-based fingerprints of the compounds were obtained using the “ChemmineR” package [53] in the R programming environment (version 4.0.3) [44], and for both analyses, heatmaps were generated for visualization within the GraphPad Prism software version 10.0.2 (232) for Windows from GraphPad Software, San Diego, CA, USA, at http://www.graphpad.com.

### 2.7. System Preparation and Molecular Dynamics Simulation Protocol

Eleven proteins’ 3D structural conformations were obtained in the manner previously described. In contrast, the 3D structure conformation of five compounds was edited and optimized using the semi-empirical General AMBER Force Field (GAFF) method [54]. In contrast, for optimizing the geometries, the Avogadro 1.2.0 program [55] was used. Proteins with ligands were prepared by molecular docking carried out in the Dockthor server (https://dockthor.lncc.br/v2/, accessed on 5 June 2023) [56,57]. Then, each system was prepared through the CHARMM-GUI [58,59,60] tool using the Solution Builder module. The molecular dynamics (MD) simulations were carried out in the Gromacs 2023 software package [61], and the CHARMM36 [62] force field was used to describe all energetic parameters for intermolecular and intramolecular interactions. A thousand steps with the steep-descent algorithm initially minimized the study systems. Subsequently, the equilibrium simulation in canonical ensemble (NVT) at 300 K by 10 ns and the MD simulation of the production in the isothermal-isobaric assembly (NPT) at 300 K at 1 bar of pressure for 100 ns were carried out. The binding energy analysis was determined using the MM/PBSA and MM/GBSA methods based on AMBER’s MMPBSA.py [63], aiming to perform end-state free energy calculations using trajectories of GROMACS.

The binding free energy (ΔG) value in the MM/PBSA (Molecular Mechanics/Poisson–Boltzmann Surface Area) is calculated based on Equation (Equation 1):(1)$$\Delta G_{bind}=\Delta E_{MM}+\Delta G_{solv}-T\Delta S$$
where:

Δ$G_{bind}$ is the binding free energy. Δ$E_{MM}$ is the molecular mechanics energy difference between the bound and unbound states of the protein-ligand complex. Δ$G_{solv}$ is the change in solvation free energy upon binding, which is further decomposed into polar ($\Delta G_{polar}$) and nonpolar ($\Delta G_{nonpolar}$) components:$\Delta G_{solv}$ = $\Delta G_{polar}$+$\Delta G_{nonpolar}$$\Delta G_{polar}$ is the polar solvation free energy and is often calculated using the Poisson–Boltzmann equation.$\Delta G_{nonpolar}$ is the nonpolar solvation free energy and is often estimated based on solvent-accessible surface area (SASA) calculations.

ΔS represents the change in entropy, with *T* being the temperature and ΔS being the change in entropy.

The binding free energy (ΔG) in the MM/GBSA (Molecular Mechanics/Generalized Born Surface Area) method is typically calculated using Equation (Equation 2):(2)$$\Delta G_{bind}=\Delta E_{MM}+\Delta G_{GB}+\Delta G_{SA}-T\Delta S$$
where:

$\Delta E_{MM}$ is the change in molecular mechanics energy, representing the gas-phase energy of the system. This includes terms for bonded and non-bonded interactions within the biomolecular complex.

$\Delta G_{GB}$ is the change in solvation free energy calculated using the Generalized Born (GB) model. It takes into account the electrostatic contributions to solvation.

$\Delta G_{SA}$ is the change in solvation free energy due to the solvent-accessible surface area (SASA). This term accounts for the nonpolar contributions to solvation.

ΔS represents the change in entropy, with *T* being the temperature and ΔS being the change in entropy.

MM/GBSA is often employed in the study of molecular interactions, such as protein-ligand binding, to estimate the thermodynamic parameters associated with these processes.

## 3. Results

### 3.1. Computational Analysis of the Purine Salvage Pathway

Although 20 species are known to circulate in the Americas [64], only the most frequent ones—*L. braziliensis*, *L. infantum*, *L. panamensis*, *L. guyanensis*, *L. mexicana*, and *L. amazonensis* were used in this study’s phylogenetic analysis of the amino acid sequences of APRT from *L. infantum*. The results showed that only *L. braziliensis* sequences were related to the APRT from *L. infantum* (Figure 1A). As advances in network biology have shown that single-protein targeting is ineffectual in treating complex illnesses, it is crucial to understand how well the proteins interfere with the operation of the intricate regulatory machinery [65,66]; APRT (ID: A4I1V1) and Adenylosuccinate lyase protein (ID: A4HS40) showed the highest centrality scores on the network whereas the putative AMP deaminase (ID: A4I876) and the AMP deaminase (ID: A4IC17) proteins showed similar centrality scores, showing their importance on the pathway (Figure 1B). Furthermore, the 11 sequences of the purine salvage pathway were also subjected to a BLASTp analysis of sequence homology toward human proteins, and the results revealed that 44 human proteins shared varying degrees of homology with the targets of *L. infantum*. Adenylosuccinate lyase protein (ID: A4HS40) and APRT (ID: A4I1V1) were the only targets with homology scores that fell below the predetermined cut-off values for homology (Figure 1C).

### 3.2. Data Collection and Virtual Screening

A search for NPs with antileishmanial and anti-APRT activities was conducted in the NuBBEDB, and the results included 33 NPs with antileishmanial activity, 10 of which have also been identified as APRT activity inhibitors. Two of the NPs described—namely Skimmianine (NuBBE_599) and Isopimpinellin (NuBBE_1280)—were isolated from *Adiscanthus fusciorus* species [67], while 8 NPs—namely 2α,3α-dihydroxyolean-12-en-28-oic acid (NuBBE_1203), 2α,3α,19α-trihydroxyurs-12-en-28-oic acid (NuBBE_1204), Chrysoeriol (NuBBE_1205), Acacetin (NuBBE_1206), 3-O-methylquercetin (NuBBE_1223), 3,3-O-dimethylquercetin (NuBBE_1224), 3,7,4-O-trimethylkaempferol (NuBBE_1225), and Penduletin 4-O-methyl ether (NuBBE_1226)—were isolated from *Vitex polygama Cham* species [68]. A search on the SwissSimilarity server using the commercial zinc-drug-like compound library was conducted to find structural analogs to the eight NPs chosen. The search produced 400 analogs for each NP, but due to levels of analog redundancy, duplicate compounds were eliminated, leaving 2460 individual compounds. Figure 2A shows the distribution of the chemical space of the dataset regarding the physicochemical properties and the drug-likeness score, while Figure 2B,C show the selection of compounds from the dataset with no violations of the Rule of Five and no potential toxicities predicted, respectively: leaving 428 individual compounds. Seven of the identified compounds exhibited the ability to bind to multiple targets in the virtual screening analysis against the 11 purine salvage pathway proteins, which are presented in Figure 2D. However, when clustering the compounds by the Tanimoto similarity score, five of them, namely 5,6,7-trimethoxy-1H-indole-2-carboxamide (PubChem CID: 4777887), 3-[Butyl(methyl)amino]-4-hydroxychromen-2-one (PubChem CID: 54733114), N-(2-methoxyethyl)-2-(7-methoxy-1H-indol-1-yl)acetamide (PubChem CID: 39315359), N,N-diethyl-4-methoxy-1-benzofuran-6-carboxamide (PubChem CID: 71688722), and 4-ethoxy-N,N-diethyl-1-benzofuran-6-carboxamide (PubChem CID: 71688752) were considered unique compounds (Figure 2E).

### 3.3. RMSD, RG, and SASA Calculations from Molecular Dynamics Simulation

Figure 3 shows the configuration of 11 *L. infantum* targets, where the conformation is shown in 3D, and a label identifies each target. In addition, it was found that the templates for the structural modeling of two models were based on crystals obtained by X-rays. In comparison, the next ten models were by Alphafold through SWISS-MODEL. Details are presented in Appendix A. Likewise, Figure 4 shows the top five compounds discovered by virtual screening. The study of the root-mean-square deviation (RMSD), which provided an average value for the conformational changes the protein underwent during the simulation time, was used to calculate the deviation of each spatial coordinate of the protein during that time. The radius of gyration (RG), which specifies how the cross-sectional area or mass distribution is spread around its central axis, was also assessed. When the RG is calculated using an advanced computational technique, the compactness of a protein is directly related to the rate at which it folds [69]. The solvent-accessible surface area (SASA) analyzes the surface area of proteins that solvent molecules can access. When a protein is under external stress, such as when it binds to a foreign material (a drug), it undergoes conformational changes that make hydrophobic residues more soluble in water and other solvents [70]. In general, the average RG and SASA values in this study indicated that the proteins linked to the medicines underwent a minor conformational shift, which might have resulted from the ligand being present in their active center. Appendix A shows the average RMSD, RG, and SASA values. Also, the Appendix A presents the different diagrams for each analyzed target. In the case of APRT, we observed that the binding to four of the five compounds stabilized the structure with an average RMSD lower than that of the APRT without ligand, while 3-[Butyl(methyl)amino]-4-hydroxychromen-2-one was the only compound that changes the structure of APRT with an average RMSD value of 0.34 nm. Also, the average RG value for APRT with 5,6,7-trimethoxy-1H-indole-2-carboxamide, N,N-diethyl-4-methoxy-1-benzofuran-6-carboxamide, and 4-ethoxy-N,N-diethyl-1-benzofuran-6-carboxamide was less than APRT without ligand. The N-(2-methoxyethyl)-2-(7-methoxy-1H-indol-1-yl) acetamide and 3-[Butyl(methyl)amino]-4-hydroxychromen-2-one compounds obtained a higher RG value, which indicates a lower compaction of the APRT structure. Compared to the other compounds, the average SASA value of APRT/3-[Butyl(methyl)amino]-4-hydroxychromen-2-one was high.

ADL/3-[Butyl(methyl)amino]-4-hydroxychromen-2-one complex presented the highest average RMSD value, while 5,6,7-trimethoxy-1H-indole-2-carboxamide was the one that improved the structural stability of ADL. The average value of the compaction analyzed with the RG shows us that the N,N-diethyl-4-methoxy-1-benzofuran-6-carboxamide compound improves the compaction of ADL and the 3-[Butyl(methyl)amino]-4-hydroxychromen-2-one and 4-ethoxy-N,N-diethyl-1-benzofuran-6-carboxamide compounds reduce the compaction of ADL. At the same time, the average value of SASA shows us that 5,6,7-trimethoxy-1H-indole-2-carboxamide was the lowest value for 3-[Butyl(methyl)amino]-4-hydroxychromen-2-one. Therefore, ADL is altered by 3-[Butyl(methyl)amino]-4-hydroxychromen-2-one binding.

### 3.4. Analysis of Protein-Ligand Binding Affinities with MM/PBSA and MM/GBSA

With the use of continuum solvation implicit models and the Molecular Mechanics Poisson–Boltzmann Surface Area (MM/PBSA) and Molecular Mechanics Generalized Born Surface Area (MM/GBSA) techniques, the binding free energy (ΔG) may be estimated. By examining several conformations acquired from the last 100 frames of the MD simulations, the ΔG of these data were ascertained. Based on the results in Appendix A, we noted that our work’s MM/PBSA technique might have done better than MM/GBSA. This could be explained by the MM/PBSA model being more sensitive to the selection of parameters. The MM/GBSA approach was therefore used in this work due to its high precision, resilience, and affordability of computational resources. The values in Appendix A show us that van der Waals energies contributed energy. Finally, the results obtained for APRT/N,N-diethyl-4-methoxy-1-benzofuran-6-carboxamide, and ADL/N,N-diethyl-4-methoxy-1-benzofuran-6-carboxamide complexes, showed the best average free energy values compared to the other systems (See Figure 5).

## 4. Discussion

An estimated 500,000 new cases of VL and 50,000 fatalities occur each year worldwide, but these numbers are believed to be underrated [71]. Children under the age of ten and those with weakened immune systems are more likely to suffer the disease than immunocompetent patients, whereas risk factors also include malnutrition, poverty, population mobility, and poor hygiene [72,73]. Because of this, the World Health Organization (WHO) has set an ambitious goal for the disease’s global eradication in its 2021–2030 neglected tropical diseases roadmap [74]. However, there are still many gaps for VL, particularly in the Americas and sub-Saharan Africa [75]. The creation of a human vaccine is still hindered by large gaps in the development pipeline, despite tremendous advances [76]; therefore, the current situation emphasizes the need for more sensitive and rapid diagnostic tests development for early detection, better treatment accessibility, and the discovery of more effective medications [77,78]. However, repurposed drugs make up most of the currently available medication options for treating VL, and in recent decades, a few novel treatment candidates have advanced through clinical trials [79]. Academia and the pharmaceutical industry are aware of the potential uses of NPs as therapeutic medications. Herewith, there has been a recent upsurge in studies to determine their effectiveness as chemotherapeutic agents for the treatment of leishmaniasis [80], while several NP groups, including quinones, terpenoids, alkaloids, coumarins, flavonoids, lignans, and neolignans, have shown antileishmanial activity [81]. Effective treatment for complex disorders, such as VL, often involves modulating numerous targets, and because of their advantageous structures, natural products offer a special chance for the development of multi-targeting medications [82]. In light of this, the current study aimed to use CADD approaches to select analogs to NPs with established antileishmanial and anti-APRT activities to target multiple PSP proteins. The compound N,N-diethyl-4-methoxy-1-benzofuran-6-carboxamide (ZINC72240060, PubChem CID: 71688722), an analog of the alkaloid Skimmianine, showed a favorable binding affinity to *L. infantum*-APRT and ADL, with no predicted toxicity and potential for oral route administration.

The computational NP repositioning method applied herein relies on the chemical structure and molecule information approach, where the structural similarity is combined with molecular activity and additional biological information to find novel relationships [83]. Likewise, as far as we are aware, no studies on the possible pharmacological activity of N,N-diethyl-4-methoxy-1-benzofuran-6-carboxamide have been published. Conversely, Skimmianine has been extensively studied for antimicrobial, antitrypanosomal, anti-insect, antiplatelet, antidiabetic, antiviral, cholinesterase inhibitory, analgesic, cardiovascular, cytotoxicity, and anti-inflammatory activities [84,85]; whereas also has shown *in vitro* activity against *L. braziliensis*, *L. amazonensis* and *L. tropica* [86,87,88]. Furthermore, its use in drug development has been justified by its pharmacological characteristics. However, several topics necessitate more research, including the intricate mechanism of action, the connection between structure and activity, toxicological information, and clinical studies [85]. The PSP analysis’s findings confirm that cheminformatics and bioinformatics offer enticing alternatives to traditional methods for identifying possible therapeutic targets [89]. These *in silico* methods enable the filtering of proteins that are highly conserved, specific, and/or selective among parasite species and strains, which is important in antileishmanial drug discovery since *in vitro* evidence of interspecies variations in the susceptibility of parasites to antileishmanial drugs has been reported [90]. The phylogenetic analysis of *L. infantum*-APRT shows that its sole related species is *L. braziliensis*-APRT when compared to other circulating species in the Americas, which restricts its use for interspecies drug development. Also, the findings further emphasize the significance of the ADL as a potential drug target because of its limited homology to human proteins and its centrality in the PSP proteins network interaction. Also, based on experimental findings, ADL is the only purine-metabolizing enzyme that is essential to the parasite life cycle on its own [91,92]. Furthermore, studies have demonstrated that whereas XPRT and HGPRT are required for the parasite, neither is required on its own [93,94]; Nevertheless, our results showed that both targets’ centrality scores on PSP were low, and they shared similarities with human proteins, characterizing them as not desirable targets. In the field of drug discovery against NTDs, the creation and design of a single chemical entity that operates at several molecular targets concurrently is receiving much attention [95]. With encouraging outcomes, CADD techniques were used to make it easier to test novel drugs with multitarget properties against illnesses including Dengue fever [96] and Chagas disease [97]. However, regarding VL, Xyloguyelline, an aporphynic alkaloid, was chosen as a possible multitarget molecule for VL treatment because it showed action against five key enzymes [98] while obtaining drug-like molecules against VL, the ZINC15 library of biogenic chemicals was used, founded on molecular docking using two targets’ binding affinities [99]. Also, Pseudoguaiacanolides and Germanocrolide, two sesquiterpenes, were chosen exclusively by computational methods that examined their capacity to bind to a range of enzyme targets [100]. The creation of novel therapeutics and pharmacology are witnessing a growing trend in multitarget medications. With the advancement of in silico, in vitro, and combination screening methodologies, the search for safer, more effective, and patient-compliant drugs will be sped up [101].

## 5. Conclusions

This work used computational analysis of available data and database research on natural products to identify a chemical that has a structural similarity to natural products with demonstrated effectiveness against *Leishmania* spp. The compound N,N-diethyl-4-methoxy-1-benzofuran-6-carboxamide (ZINC72240060, PubChem CID: 71688722), an analog of the alkaloid Skimmianine, showed a favorable binding affinity to *L. infantum*-APRT and ADL, with no predicted toxicity and potential for oral route administration. The findings of this work support the possibility of *in vitro* and *in vivo* research employing the selected compound to validate its potential as a treatment candidate for VL.

## Figures and Tables

**Figure 1 tropicalmed-09-00041-f001:**
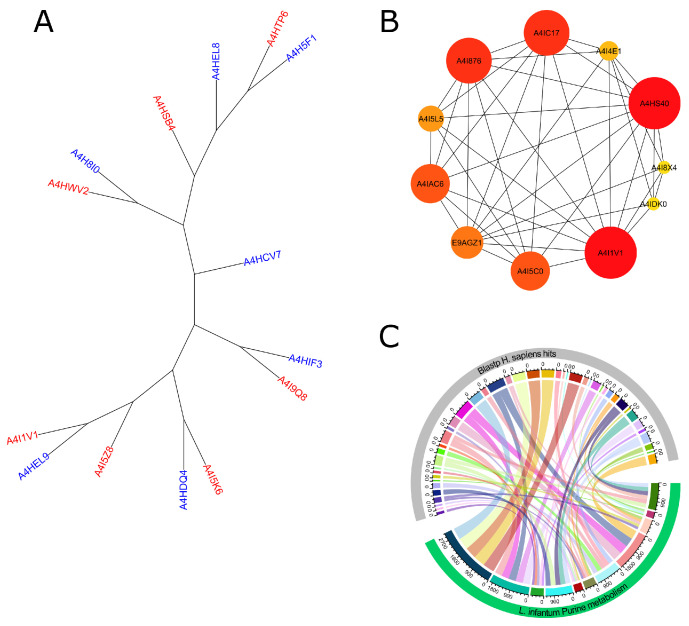
Computational analysis of the purine salvage pathway. Leishmanial homologs and the *L. infantum* APRT proteins’ phylogenetic tree. Sequences of *L. infantum* are represented by the blue labels, whereas those of *L. braziliensis* are represented by the red labels (**A**). Network diagram of the CytoHubba purine salvage pathway, where redder and larger nodes indicate a higher degree of centrality. Where Adenine phosphoribosyltransferase (ID: A4I1V1), Adenylosuccinate lyase (ID: A4HS40), Putative AMP deaminase (ID: A4I876), AMP deaminase (ID: A4IC17), Guanine deaminase (ID: A4I4E1), GMP synthase (ID: E9AGZ1), Adenylate kinase isoenzyme (ID: A4I5L5), Guanylate kinase-like protein (ID: A4IDK0), Adenosine kinase (ID: A4I5C0), and Adenosine kinase (ID: A4IAC6) (**B**). Sequence homology analysis, whereas the chord plots display the BLASTp results (**C**).

**Figure 2 tropicalmed-09-00041-f002:**
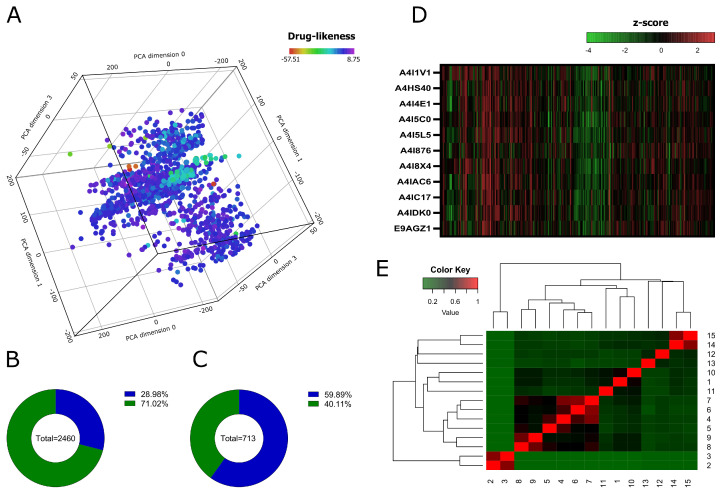
Molecular properties analysis and virtual screening. The chemical space of the generated dataset is represented visually by 3D-PCA (**A**). Pie charts show the analysis of Lipinski’s rule of five (**B**) and predicted toxicities (**C**) from the dataset. Heatmap showing the z-scores of the binding affinities of 428 compounds against the protein targets of the purine salvage pathway (**D**). Heatmap generated with Tanimoto scoring matrix of similar structures among compounds (**E**).

**Figure 3 tropicalmed-09-00041-f003:**
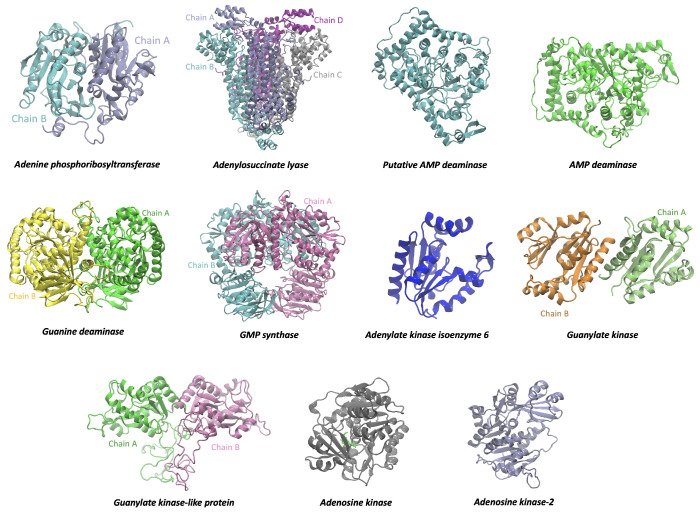
Targets. 3D protein structure of 11 *L. infantum* targets obtained by homology modeling. Cofactors, metal ions, and the presence of multiple protein monomers were considered.

**Figure 4 tropicalmed-09-00041-f004:**
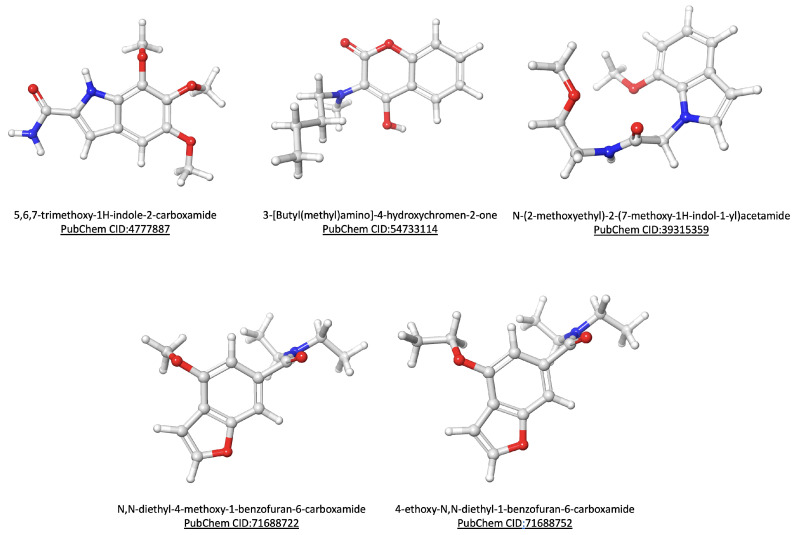
Ligands. The chemical structure of the selected compounds was analyzed by virtual screening.

**Figure 5 tropicalmed-09-00041-f005:**
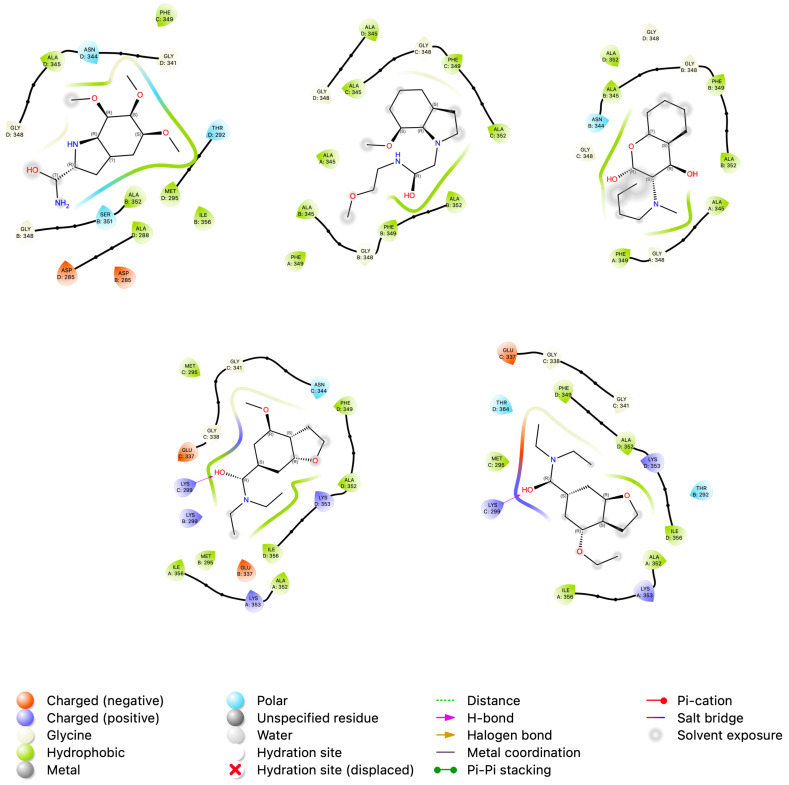
Pictorial 2D representation. The best binding free energy between Adenylosuccinate lyase and compounds was computed from the last 10 ns (100 frames) of 100 ns MD simulation.

## Data Availability

Data are contained in Appendix A within the article.

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
