# Peer review of "Targeting with Structural Analogs of Natural Products the Purine Salvage Pathway in Leishmania (Leishmania) infantum by Computer-Aided Drug-Design Approaches"

_tropicalmed, 2024, doi:10.3390/tropicalmed9020041_

Round 1

Reviewer 1 Report

Comments and Suggestions for Authors

Authors present an important study to identify structural analogs of natural products with the potential to treat visceral leishmaniasis. Computational analysis of available data and database research led to the identification of the compound, an analog of the alkaloid Skimmianine, that has a potential of anti-leishmanial activity. The manuscript requires the minor revisions, for example, not all latin names are in italics and it needs to be corrected. 

Comments on the Quality of English Language

Minor English revisions are required.

Author Response

Authors present an important study to identify structural analogs of natural products with the potential to treat visceral leishmaniasis. Computational analysis of available data and database research led to the identification of the compound, an analog of the alkaloid Skimmianine, that has a potential of anti-leishmanial activity. The manuscript requires the minor revisions, for example, not all latin names are in italics and it needs to be corrected. (Minor English revisions are required.).

ANSWER: We thank the reviewer for your kind and supportive comments. We have worked thoroughly to improve the manuscript’s quality and correct missing details.

Reviewer 2 Report

Comments and Suggestions for Authors

Barazorda-Ccahuana et al., present a virtual screening of naturak products derivatives as putative anti-leishmanial compounds. The work can be of high interest and provide a good starting point for further study. Still,I have some concerns and comments that could benefit the manuscript:

Minor

There are some instances in text where leishmania species are not in italics, revise.

Some labels are not legible enough in Figure 1B

Figures 3,4 & 5 are not correctly placed, considering their mention in main text.

2D-structures in Figure 5 are wrongly displayed, this would be the case if docking results were loaded to Maestro as PDB files. Please make a careful revision.

Line 177: "the equilibrium simulation in canonical assembly"; assembly is not the correct term.

The authors do not provide details on the continuum model for GBSA calculations.

The authors used SWISS-model server to carry out homology modelling. First, I must state that I hold nothing against this in itself. However, with the advent of AI powered algorithms such as Alphafold, some  may question the omision of these methods. Could the authors provide a comment on this?

Major

Considering that most of the computational work relies on homology modelling the authors provide little proof on the quality and overall pertinence of the obtained models. Please include this information as part of the supplementary material.

Based on the main text, it is implied that only one simulation per complex was carried out. This is quite troublesome in the homology modelling context. More so considering some of the trends observed in RMSD and RG, additionally, the authors should include RMSF plots to further illustrate the overall stability of the homologued proteins. Plus, and I must say this is entirely subjective, the overall stability of models should be assessed with longer simulations, just for the free protein at least.

On a similar note, docking protocols in homology models can be difficult in some cases. Thus, it would be helpful to discussed the rationale (if any) to improve this assessment. I mention this considering the values retrieved by endpoint methods. Most of them are rather discrete or positive even, such trend and some of the observed plots in RMSD and RG seem to suggest that selected binding modes (or perhaps protein conformation) is not reasonable as starting point. Summarizing, the authors should double check their docking protocol and consider additional binding poses not just "the best" score.

Finally, if the work is only theoretical in nature the authors should improve the discussion on the viability of the identified compounds (considering that result may be subject to change, based on my previous comments). Indeed, chemography and chemical space delimitation are useful tools but little is done to discuss protein-ligand interactions nor their significance.

Comments on the Quality of English Language

English quality is mostly ok, I suggest a careful revision and minor style edits.

Author Response

Barazorda-Ccahuana et al., present a virtual screening of natural products derivatives as putative anti-leishmanial compounds. The work can be of high interest and provide a good starting point for further study. Still, I have some concerns and comments that could benefit the manuscript:

ANSWER: We thank the reviewer for your kind and supportive comments. We have worked thoroughly to improve the manuscript’s quality and correct missing details.

Minor

There are some instances in text where leishmania species are not in italics, revise.

ANSWER: Thank you for your assessment. We have corrected the error.

Some labels are not legible enough in Figure 1B

ANSWER: Thank you for the recommendation. Since the labels overlapped the network lines as their size expanded, we decided to remain with the current figure.

Figures 3,4 & 5 are not correctly placed, considering their mention in main text.

ANSWER: Thank you for your assessment. We have corrected the error.

2D-structures in Figure 5 are wrongly displayed, this would be the case if docking results were loaded to Maestro as PDB files. Please make a careful revision.

ANSWER: Thank you for your assessment. However, the models were acquired after the protein-ligand complex's MDS; the final frame was then considered and turned into a PDB file. Finally, this was loaded into Maestro and the 2D representation was obtained.

Line 177: "the equilibrium simulation in canonical assembly"; assembly is not the correct term.

ANSWER: Thank you for your assessment. We have corrected the error.

The authors do not provide details on the continuum model for GBSA calculations.

ANSWER: Thank you for your assessment. We have added the missing information.

The authors used SWISS-model server to carry out homology modelling. First, I must state that I hold nothing against this in itself. However, with the advent of AI powered algorithms such as Alphafold, some may question the omision of these methods. Could the authors provide a comment on this?

ANSWER: We appreciate your assessment and acknowledge your concerns. However, some recent studies, such as Scardino et al. [“How good are AlphaFold models for docking-based virtual screening?.” iScience vol. 26,1 105920. 30 Dec. 2022, doi:10.1016/j.isci.2022.105920] has pointed out AlphaFold’s inability to generate models reliable to be used for high throughput docking and post-modeling refinement strategies are recommended. In addition, SwissModel uses high-quality models from the AlphaFold DB, as mentioned on the Swissmodel server page (https://swissmodel.expasy.org/docs/repository help) as quality controls. In light of this, we have included the specifics of each model in Table S1 of the Supplementary and updated the manuscript with these details.

Major

Considering that most of the computational work relies on homology modelling the authors provide little proof on the quality and overall pertinence of the obtained models. Please include this information as part of the supplementary material.

ANSWER: Thank you for your assessment. As stated earlier we have added this information to the manuscript.

Based on the main text, it is implied that only one simulation per complex was carried out. This is quite troublesome in the homology modelling context. More so considering some of the trends observed in RMSD and RG, additionally, the authors should include RMSF plots to further illustrate the overall stability of the homologued proteins. Plus, and I must say this is entirely subjective, the overall stability of models should be assessed with longer simulations, just for the free protein at least.

ANSWER: Thank you for your assessment. However, this information is already shown in the supplementary figures S1 to S11.

On a similar note, docking protocols in homology models can be difficult in some cases. Thus, it would be helpful to discussed the rationale (if any) to improve this assessment. I mention this considering the values retrieved by endpoint methods. Most of them are rather discrete or positive even, such trend and some of the observed plots in RMSD and RG seem to suggest that selected binding modes (or perhaps protein conformation) is not reasonable as starting point. Summarizing, the authors should double check their docking protocol and consider additional binding poses not just "the best" score.

ANSWER: We appreciate your assessment and acknowledge your concerns. Since docking scores are estimates of interaction energies, they are not suitable for use in an exact hit ranking. Sorting through the many molecules that are unlikely to bind from those that are likely to bind to two or more targets was the main objective of the molecular docking approach. Additionally, by employing the molecular dynamics protocol of the complex in a solvated environment, we were able to see the dynamic behavior of the receptor and ligand. By applying both strategies in order, we are forced to choose just one of the 428 compounds that meet the necessary criteria, making this a logical strategy. In addition, the study's protocol—which was modified slightly for each case—has already been published [Barazorda-Ccahuana, Haruna Luz et al. “Computer-aided drug design approaches applied to screen natural product's structural analogs targeting arginase in Leishmania spp.” F1000Research vol. 12 93. 13 Jul. 2023, doi:10.12688/f1000research.129943.2; Goyzueta-Mamani, Luis Daniel et al. “In Silico Analysis of Metabolites from Peruvian Native Plants as Potential Therapeutics against Alzheimer's Disease.” Molecules (Basel, Switzerland) vol. 27,3 918. 28 Jan. 2022, doi:10.3390/molecules27030918; Goyzueta-Mamani, Luis Daniel et al. “Antiviral Activity of Metabolites from Peruvian Plants against SARS-CoV-2: An In Silico Approach.” Molecules (Basel, Switzerland) vol. 26,13 3882. 25 Jun. 2021, doi:10.3390/molecules26133882].

Finally, if the work is only theoretical in nature the authors should improve the discussion on the viability of the identified compounds (considering that result may be subject to change, based on my previous comments). Indeed, chemography and chemical space delimitation are useful tools but little is done to discuss protein-ligand interactions nor their significance. (English quality is mostly ok, I suggest a careful revision and minor style edits.)

ANSWER: We recognize your concerns and value your assessment. As previously said, it appears that the workflow in use uses a series of bioinformatic approaches sequentially to choose compounds that can bind to several targets and that those targets also need to exhibit ideal qualities. Because our study's conclusions did not solely depend on binding affinities, we did not think it necessary to examine them in detail. Also, as discussed in the manuscript, the repositioning method used in the analysis is based on the chemical structure and molecule information approach, in which novel associations are found by combining structural similarities with molecular activity and additional biological information; and we believe that the fact that Skimmianine, the substance that shares structural similarity with the recently reported novel molecule, has been well studied for its pharmacological activity and has shown antileishmanial qualities in vitro, already serves as evidence for our findings.

Reviewer 3 Report

Comments and Suggestions for Authors

Leishmania lacks de novo purine biosynthesis and are therefore dependent on purine salvage. In this manuscript, an in silico approach to target purine salvage is introduced. Because there are many backup systems in purine salvage, a main challenge is to find drugs that hit multiple targets and cannot be circumvented. I found the manuscript interesting but felt some information was lacking in the introduction to grasp the thinking behind the study. First of all, I was uncertain what compounds were selected for the approach. It is mentioned that structural analogues of natural products that shows anti-leishmanial and anti-APRT properties were selected for virtual screening. However, there is no reference which natural product analogues are used and how it is known that these compounds have anti-leishmanial properties and that they inhibit APRT. It is also unclear if they have been tested against Leishmania APRT or if they had been tested on APRT from other species.  It is also unclear from the introduction why APRT was chosen as the initial target. It has previously been proposed that the adenine is primarily salvaged by adenine amidohydrolase, a Leishmania-specific enzyme that converts adenine to hypoxanthine (see reference 86 in the manuscript), which in turn can be salvaged by XPRT or HGPRT. I think it is important to mention this pathway in the introduction, and stress more clearly in the introduction that the idea behind the approach presented is to find compounds that hit multiple targets and thereby circumvent this problem.

Other issues:

1. It is not clear from the abstract that the incentive was to start with anti-APRT agents but that the goal was to find drugs that hit multiple targets.

2. Include a review about the nucleotide metabolism in Leishmania (or trypanosomatids in general) in the introduction.

3. Row 97. Correct “homologous” (should be homologues)

4. Row 145. Some purine salvage enzymes are missing in the virtual screening such as adenine amidohydrolase, XPRTase, HGPRTase, all the purine nucleoside cleavage enzymes (IAG-NH, IG-NH, MTPase) ADSS and other adenylate isozymes than the one mentioned. Give an explanation how it was decided which enzymes to include and not.

5. Row 252-256. Are there any references to the following two statements?

Statement 1: “When the RG is calculated using an advanced computational technique, the compactness of a protein is directly related to the rate at which it folds.”  

Statement 2: “When a protein is under external stress, such as when it binds to a foreign material (a drug), it undergoes conformational changes that make hydrophobic residues more soluble in water and other solvents.”

6. I think that an interesting finding was that the drug also hits ADL that is mentioned as the only essential enzyme in purine salvage. I think this enzyme should be mentioned in the abstract as well. Can you say anything about which of the two enzymes Skimmianine is likely to have the strongest interaction with? It would also be interesting to have more discussion about the two enzymes. It has been proposed that ADL inhibition leads to that IMP formed by salvage of hypoxanthine get trapped as adenylosuccinate in the trypanosomatids and that this could be the reason why they are more vulnerable to inhibition of this enzyme than to ADSS. An interesting consequence is that if both APRT and ADL are inhibited, it will then interfere with the salvage of adenine as well as hypoxanthine (and thereby also the hypoxanthine obtained from adenine via adenine amidohydrolase).

Comments on the Quality of English Language

I think the English was good and the only general issue is that the paragraphs tended to be very long. An example of that is that all the introduction is in a single paragraph.

Minor correction:

Row 97. Correct “homologous” (should be homologues)

Author Response

Leishmania lacks de novo purine biosynthesis and are therefore dependent on purine salvage. In this manuscript, an in silico approach to target purine salvage is introduced. Because there are many backup systems in purine salvage, a main challenge is to find drugs that hit multiple targets and cannot be circumvented. I found the manuscript interesting but felt some information was lacking in the introduction to grasp the thinking behind the study. First of all, I was uncertain what compounds were selected for the approach. It is mentioned that structural analogues of natural products that shows anti-leishmanial and anti-APRT properties were selected for virtual screening. However, there is no reference which natural product analogues are used and how it is known that these compounds have anti-leishmanial properties and that they inhibit APRT. It is also unclear if they have been tested against Leishmania APRT or if they had been tested on APRT from other species.  It is also unclear from the introduction why APRT was chosen as the initial target. It has previously been proposed that the adenine is primarily salvaged by adenine amidohydrolase, a Leishmania-specific enzyme that converts adenine to hypoxanthine (see reference 86 in the manuscript), which in turn can be salvaged by XPRT or HGPRT. I think it is important to mention this pathway in the introduction, and stress more clearly in the introduction that the idea behind the approach presented is to find compounds that hit multiple targets and thereby circumvent this problem.

ANSWER: We thank the reviewer for your kind and acknowledge your concerns. As mentioned in the manuscript, the NPs were chosen using the NuBBEDB database, a curated collection of Brazilian NPs that lists the compounds according to the pharmacological characteristics that have been described in the literature. Ten of the 33 NPs that were listed as antileishmanial were tested against APRT, and the actions of the compounds were detailed in two publications that the manuscript cites. Then, the SwissSimilarity server was used to find the structural analogs, which after removing duplicates, were then found in 2460 distinct compounds. These compounds were filtered again, for their oral absorption and toxicity profile, resulting in 428 that were used in the virtual screening analysis. Besides that, we have included your recommendations in the on the manuscript.

Other issues:

It is not clear from the abstract that the incentive was to start with anti-APRT agents but that the goal was to find drugs that hit multiple targets.

ANSWER: Thank you for your assessment. We have added the missing information.

Include a review about the nucleotide metabolism in Leishmania (or trypanosomatids in general) in the introduction.

ANSWER: We appreciate your evaluation. Nonetheless, we believe that the six paragraphs that make up the introduction part are a good length and cover all the major points of the work. In addition to being a substantial point, we believe that the introduction's structure might be broken by including a paragraph summarizing this subject. However, when biochemically confirming these findings, it is a topic that has to be addressed and discussed.

Row 97. Correct “homologous” (should be homologues)

ANSWER: Thank you for your assessment. We have corrected the error.

Row 145. Some purine salvage enzymes are missing in the virtual screening such as adenine amidohydrolase, XPRTase, HGPRTase, all the purine nucleoside cleavage enzymes (IAG-NH, IG-NH, MTPase) ADSS and other adenylate isozymes than the one mentioned. Give an explanation how it was decided which enzymes to include and not.

ANSWER: Thank you for your evaluation. Nevertheless, as indicated in the manuscript, the protein targets examined in this investigation are those that were obtained from the STRING database and are related to the molecular network of L. infantum-APRT.

Row 252-256. Are there any references to the following two statements?

Statement 1: “When the RG is calculated using an advanced computational technique, the compactness of a protein is directly related to the rate at which it folds.”

ANSWER: We appreciate your evaluation. For this sentence, we don't think using a specific reference is necessary.

Statement 2: “When a protein is under external stress, such as when it binds to a foreign material (a drug), it undergoes conformational changes that make hydrophobic residues more soluble in water and other solvents.”

ANSWER: We appreciate your evaluation. For this sentence, we don't think using a specific reference is necessary.

I think that an interesting finding was that the drug also hits ADL that is mentioned as the only essential enzyme in purine salvage. I think this enzyme should be mentioned in the abstract as well. Can you say anything about which of the two enzymes Skimmianine is likely to have the strongest interaction with? It would also be interesting to have more discussion about the two enzymes. It has been proposed that ADL inhibition leads to that IMP formed by salvage of hypoxanthine get trapped as adenylosuccinate in the trypanosomatids and that this could be the reason why they are more vulnerable to inhibition of this enzyme than to ADSS. An interesting consequence is that if both APRT and ADL are inhibited, it will then interfere with the salvage of adenine as well as hypoxanthine (and thereby also the hypoxanthine obtained from adenine via adenine amidohydrolase). (I think the English was good and the only general issue is that the paragraphs tended to be very long. An example of that is that all the introduction is in a single paragraph.) Minor correction: Row 97. Correct “homologous” (should be homologues)

ANSWER: We appreciate your evaluation. We have added information regarding ADL in the abstract. Skimmianine has higher binding affinities toward APRT; Not only is the suggestion intriguing, but we believe that discussing the enzymes would be more fitting for a biochemical validation paper than the kind we are talking about here, which is about in silico target and drug development. We have worked thoroughly to improve the manuscript’s quality and correct missing details.

Round 2

Reviewer 2 Report

Comments and Suggestions for Authors

Most of my concerns have been addressed.

Regarding Figure 5, I thank the authors for the clarification, still the issue is the representation is wrong due to the missing bond orders. This shall be fixed by loading the frame in PDB and using the preprocess module in protein preparation wizard.

I still think that discussion on protein-ligand contacts should be included, at least as concurrences diagrams or something similar to assess putative metastability on the binding modes.

Comments on the Quality of English Language

English quality is mostly ok

Author Response

Most of my concerns have been addressed.

ANSWER: We thank the reviewer for your kind and supportive comments. We have worked thoroughly to improve the manuscript’s quality and correct missing details.

Regarding Figure 5, I thank the authors for the clarification, still the issue is the representation is wrong due to the missing bond orders. This shall be fixed by loading the frame in PDB and using the preprocess module in protein preparation wizard. I still think that discussion on protein-ligand contacts should be included, at least as concurrences diagrams or something similar to assess putative metastability on the binding modes.

ANSWER: We appreciate the reviewer's comments. We have reviewed the protein preparation Wizard module of the Maestro program, and our consideration of it seems to be unnecessary since the PDB file comes from a dynamic and optimized system. Likewise, we reviewed the module and we have not been able to observe any changes. The information regarding protein-ligand stability is already shown in the supplementary figures S1 to S11.

Reviewer 3 Report

Comments and Suggestions for Authors

A few sections have been improved, but most things still remain as they were before in the manuscript. It is therefore necessary to go through a second more thorough round of revision.

I think the response written to my main review paragraph was good but it was not implemented in the manuscript. When reading the introduction, it is still unclear where the compounds come from and I think it will help the understanding to at least have a rough idea about this in the introduction although the main description comes in Materials and methods. For the introduction it would be sufficient to add the following sentence: "The compunds were chosen using the NuBBEDB database, a curated collection of Brazilian NPs that lists the compounds according to the pharmacological characteristics".

Another thing from my first reviewing paragraph is that this section is not commented at all: "It has previously been proposed that the adenine is primarily salvaged by adenine amidohydrolase, a Leishmania-specific enzyme that converts adenine to hypoxanthine (see reference 86 in the manuscript), which in turn can be salvaged by XPRT or HGPRT. I think it is important to mention this pathway in the introduction, and stress more clearly in the introduction that the idea behind the approach presented is to find compounds that hit multiple targets and thereby circumvent this problem."

Other issues (only the ones not properly addresses are listed):

1. Include a review about the nucleotide metabolism in Leishmania (or trypanosomatids in general) in the introduction.

-As noted above, it is fine if the reference is on nucleotide metabolism in trypanosomatids in general (I am not sure if any recent one exist on Leishmania).

2. Row 97. Correct “homologous” (should be homologues)

-It seems like an additional correction has been made on row 104 (which should not have been done). Can you change the one marked in organge back to "homologous"? In this case, it is an adjective and it should therefore be "homologous".

3. Row 252-256. Are there any references to the following two statements? Statement 1: “When the RG is calculated using an advanced computational technique, the compactness of a protein is directly related to the rate at which it folds.” Statement 2: “When a protein is under external stress, such as when it binds to a foreign material (a drug), it undergoes conformational changes that make hydrophobic residues more soluble in water and other solvents.”

-Here the response was that references are not needed. However, it is now written as statements, which make the reader expect references. An alternative to use references, could be to rephrase the sentences to be less definitive (for example by saying: a possible reason could be..............).

4. The introduction is still not divided into paragraphs. One way to do it could be to split it at row 43 and 59. Then you have a first general paragraph, a second paragraph with natural products and a third paragraph for purine metabolism.

Comments on the Quality of English Language

In general good (see above for the few issues that exists).

Author Response

A few sections have been improved, but most things still remain as they were before in the manuscript. It is therefore necessary to go through a second more thorough round of revision.

ANSWER: We thank the reviewer for your kind and supportive comments. We have worked thoroughly to improve the manuscript’s quality and correct missing details.

I think the response written to my main review paragraph was good but it was not implemented in the manuscript. When reading the introduction, it is still unclear where the compounds come from and I think it will help the understanding to at least have a rough idea about this in the introduction although the main description comes in Materials and methods. For the introduction it would be sufficient to add the following sentence: "The compunds were chosen using the NuBBEDB database, a curated collection of Brazilian NPs that lists the compounds according to the pharmacological characteristics".

ANSWER: We appreciate your evaluation. The needed details have been added to the manuscript's introduction section.

Another thing from my first reviewing paragraph is that this section is not commented at all: "It has previously been proposed that the adenine is primarily salvaged by adenine amidohydrolase, a Leishmania-specific enzyme that converts adenine to hypoxanthine (see reference 86 in the manuscript), which in turn can be salvaged by XPRT or HGPRT. I think it is important to mention this pathway in the introduction, and stress more clearly in the introduction that the idea behind the approach presented is to find compounds that hit multiple targets and thereby circumvent this problem."

ANSWER: Thank you for your assessment. The manuscript's core idea has undergone modifications to be shown more clearly. Although XPRT and HGPR have been the subject of much research, our study indicates that these two targets are not suitable candidates for drug development due to their high similarity to human proteins and low centrality on the pathway under investigation. Therefore, we decided against adding the suggested paragraph to its description.

Other issues (only the ones not properly addresses are listed):

  1. Include a review about the nucleotide metabolism in Leishmania (or trypanosomatids in general) in the introduction. As noted above, it is fine if the reference is on nucleotide metabolism in trypanosomatids in general (I am not sure if any recent one exist on Leishmania).

ANSWER: Thank you for your assessment. We have added the information in the introduction section of the manuscript.

  1. Row 97. Correct “homologous” (should be homologues)

-It seems like an additional correction has been made on row 104 (which should not have been done). Can you change the one marked in orange back to "homologous"? In this case, it is an adjective and it should therefore be "homologous".

ANSWER: Thank you for your assessment. We have corrected the error.

  1. Row 252-256. Are there any references to the following two statements? Statement 1: “When the RG is calculated using an advanced computational technique, the compactness of a protein is directly related to the rate at which it folds.” Statement 2: “When a protein is under external stress, such as when it binds to a foreign material (a drug), it undergoes conformational changes that make hydrophobic residues more soluble in water and other solvents.” Here the response was that references are not needed. However, it is now written as statements, which make the reader expect references. An alternative to use references, could be to rephrase the sentences to be less definitive (for example by saying: a possible reason could be..............).

ANSWER: Thank you for your assessment. We have corrected the missing information.

  1. The introduction is still not divided into paragraphs. One way to do it could be to split it at row 43 and 59. Then you have a first general paragraph, a second paragraph with natural products and a third paragraph for purine metabolism.

ANSWER: Thank you for your assessment. We have corrected the error.

Round 3

Reviewer 3 Report

Comments and Suggestions for Authors

I think there are still issues remaining and I think it would be good if the authors can go through the points more carefully this time. This is the second time not enough of effort is spent on improving the paper itself based on the comments.

-Still nothing is mentioned in the introduction about that it has previously been proposed that the adenine is primarily salvaged by adenine amidohydrolase, a Leishmania-specific enzyme that converts adenine to hypoxanthine (see reference 86 in the manuscript). This is a key point for the whole study since it means that it probably not is enough to inhibit APRT alone.

-Reference on nucleotide metabolism. The review selected (number 23) is primarily on extracellular nucleotide metabolism but the enzymes discussed in this paper are intracellular enzymes. Take a more general review on nucleotide metabolism in trypansomatids.

-Are there any references added for the two statements (or rephrasing them)? It is at least not among the marked changes in the manuscript.

-Minor issue: on row 60 it seems to be added a few letters extra by mistake.

Comments on the Quality of English Language

It is OK.

Author Response

I think there are still issues remaining and I think it would be good if the authors can go through the points more carefully this time. This is the second time not enough of effort is spent on improving the paper itself based on the comments.

ANSWER: We thank the reviewer for your kind and supportive comments. We have worked thoroughly to improve the manuscript’s quality and correct missing details.

-Still nothing is mentioned in the introduction about that it has previously been proposed that the adenine is primarily salvaged by adenine amidohydrolase, a Leishmania-specific enzyme that converts adenine to hypoxanthine (see reference 86 in the manuscript). This is a key point for the whole study since it means that it probably not is enough to inhibit APRT alone.

ANSWER: Thank you for your assessment. We have added the missing information.

-Reference on nucleotide metabolism. The review selected (number 23) is primarily on extracellular nucleotide metabolism but the enzymes discussed in this paper are intracellular enzymes. Take a more general review on nucleotide metabolism in trypansomatids.

ANSWER: Thank you for your assessment. We have added the corrected the information.

-Are there any references added for the two statements (or rephrasing them)? It is at least not among the marked changes in the manuscript.

ANSWER: Thank you for your assessment. We have corrected the error.

-Minor issue: on row 60 it seems to be added a few letters extra by mistake.

ANSWER: Thank you for your assessment. We have corrected the error.

Round 4

Reviewer 3 Report

Comments and Suggestions for Authors

I think most of the points now are addressed but unfortunately as it is written now, the aminohydrolase part is not put in an optimal context with the rest.

The starting point of the manuscript is that compounds were selected with anti-APRT and anti-Leismanial properties. However, it is then an enigma how the compounds can be anti-Leishmanial although APRT is not essential (since adenine also can be salvaged via adenine aminohydrolase).  In order for the study to make sense it therefore needs to be stated that the aim was to find out if the compounds could have other targets as well which could explain how they can be anti-Leishmanial although APRT is not essential.  With this aim in mind, the study was successful in finding ADL as a second target, which is truly essential and could perfecly explain the anti-Leishmanial properties.

It would also be good if this aim can be included in the abstract, which is a bit vague as it is. 

Author Response

I think most of the points now are addressed but unfortunately as it is written now, the aminohydrolase part is not put in an optimal context with the rest.

ANSWER: We thank the reviewer for your comments. We have worked to improve the manuscript’s quality and correct missing details.

The starting point of the manuscript is that compounds were selected with anti-APRT and anti-Leismanial properties. However, it is then an enigma how the compounds can be anti-Leishmanial although APRT is not essential (since adenine also can be salvaged via adenine aminohydrolase).  In order for the study to make sense it therefore needs to be stated that the aim was to find out if the compounds could have other targets as well which could explain how they can be anti-Leishmanial although APRT is not essential.  With this aim in mind, the study was successful in finding ADL as a second target, which is truly essential and could perfecly explain the anti-Leishmanial properties. It would also be good if this aim can be included in the abstract, which is a bit vague as it is.

ANSWER: We thank the reviewer for your comments. We have added the required information to the abstract.
